# Comprehensive Gene Analysis of IgG4-Related Ophthalmic Disease Using RNA Sequencing

**DOI:** 10.3390/jcm9113458

**Published:** 2020-10-27

**Authors:** Masaki Asakage, Yoshihiko Usui, Naoya Nezu, Hiroyuki Shimizu, Kinya Tsubota, Kazuhiko Umazume, Naoyuki Yamakawa, Tomohiro Umezu, Hirotsugu Suwanai, Masahiko Kuroda, Hiroshi Goto

**Affiliations:** 1Department of Ophthalmology, Tokyo Medical University, 6-7-1 Nishi-shinjuku, Shinjuku-ku, Tokyo 160-0023, Japan; patty.m.best@gmail.com (M.A.); naoya.nezu@gmail.com (N.N.); sardine_harbor@yahoo.co.jp (H.S.); tsubnkin@hotmail.co.jp (K.T.); kazuhiko-uma@kvf.biglobe.ne.jp (K.U.); yamakawa@tokyo-med.ac.jp (N.Y.); goto1115@tokyo-med.ac.jp (H.G.); 2Department of Molecular Pathology, Tokyo Medical University, 6-7-1 Nishi-shinjuku, Shinjuku-ku, Tokyo 160-0023, Japan; t_umezu@tokyo-med.ac.jp (T.U.); kuroda@tokyo-med.ac.jp (M.K.); 3Department of Diabetes, Metabolism and Endocrinology, Tokyo Medical University, 6-7-1 Nishi-shinjuku Shinjuku-ku, Tokyo 160-0023, Japan; suwanai-h@umin.ac.jp

**Keywords:** IgG4-related disease, IgG4-related ophthalmic disease, RNA sequencing, MMP12, SPP1, orbital lymphoproliferative disorders

## Abstract

High-throughput RNA sequencing (RNA-seq) uses massive parallel sequencing technology, allowing the unbiased analysis of genome-wide transcription levels and tumor mutation status. Immunoglobulin G4-related ophthalmic disease (IgG4-ROD) is a fibroinflammatory disease characterized by the enlargement of the ocular adnexal tissues. We analyzed RNA expression levels via RNA-seq in the biopsy specimens of three patients diagnosed with IgG4-ROD. Mucosa-associated lymphoid tissue (MALT) lymphoma, reactive lymphoid hyperplasia (RLH), normal lacrimal gland tissue, and adjacent adipose tissue were used as the controls (*n* = 3 each). RNA-seq was performed using the NextSeq 500 system, and genes with |fold change| ≥ 2 and *p* < 0.05 relative to the controls were defined as differentially expressed genes (DEGs) in IgG4-ROD. To validate the results of RNA-seq, real-time polymerase chain reaction (PCR) was performed in 30 IgG4-ROD and 30 orbital MALT lymphoma tissue samples. RNA-seq identified 35 up-regulated genes, including matrix metallopeptidase 12 (MMP12) and secreted phosphoprotein 1 (SPP1), in IgG4-ROD tissues when compared to all the controls. Many pathways related to the immune system were included when compared to all the controls. Expressions of MMP12 and SPP1 in IgG4-ROD tissues were confirmed by real-time PCR and immunohistochemistry. In conclusion, we identified novel DEGs, including those associated with extracellular matrix degradation, fibrosis, and inflammation, in IgG4-ROD biopsy specimens. These data provide new insights into molecular pathogenetic mechanisms and may contribute to the development of new biomarkers for diagnosis and molecular targeted drugs.

## 1. Introduction

Immunoglobulin (Ig) G4-related disease (IgG4-RD) is a relatively new disease concept that was originally proposed in Japan. This disease causes fibrosis in the organs throughout the body including the eyes, and the organs become swollen [1]. Involvement of follicular T cells (Tfh) and regulatory T cells (Tregs) has been reported [2,3], but the pathogenetic mechanisms remain to be elucidated. Various etiologies, such as infectious diseases [4], have been proposed.

IgG4-related ophthalmic disease (IgG4-ROD) is an orbital lymphoproliferative disorder that show enlargement of the ocular appendages, including the lacrimal gland. IgG4-ROD is classified as a benign disease, though malignant lymphoproliferative diseases, such as mucosa-associated lymphoid tissue (MALT) lymphoma, can also occur in the orbit. Clinical symptoms, diagnostic imaging, histology, molecular analysis, and flow cytometry using biopsy specimens are the most commonly used diagnostic tools for orbital lymphoproliferative disorders [5,6]. Differentiation of these orbital lymphoproliferative disorders is often difficult, as these lesions share clinical, imaging, and histologic features, as well as molecular markers [5,7,8,9,10]. In addition, several reports have suggested that 12% of orbital MALT lymphoma result from a background of IgG4-ROD [11,12].

Differential diagnosis between IgG4-ROD and orbital MALT lymphoma, both of B-cell lineage, remains complicated and challenging in routine clinical practice because of the lack of specific diagnostic biomarkers. Elevated serum IgG4 is not sufficiently sensitive or specific enough for this purpose [13,14]. Therefore, approximately one third of IgG4-ROD patients do not completely meet the clinical criteria and are diagnosed with ‘possible’ or ‘probable’ IgG4-ROD, which contributes to clinical confusion and delayed diagnosis [6,15]. Reactive lymphoid hyperplasia (RLH) is another benign lymphoproliferative disorder. To the best of our knowledge, there is also no report on the differentiation between IgG4-ROD and RLH based on gene profiling, making differential diagnosis difficult.

RNA sequencing (RNA-seq) is a type of next-generation sequencing (NGS) technology that allows the comprehensive evaluation and quantification of all subtypes of RNA expressed in tissues [16]. Compared with the microarray analysis conventionally used in transcriptome analysis, the correlation coefficient of the fold change (FC) of the gene expression level between RNA-seq and the microarray was approximately 0.7 [17]. Moreover, RNA-seq detected expression differences in some genes that were not significantly different in the microarray, and the differences in the expression of these genes were confirmed by a quantitative polymerase chain reaction (PCR) analysis [17]. Other advantages of RNA-seq include a greater dynamic range and the ability to identify abnormally altered genes or molecular pathways that may lead to the discovery of novel diagnostic biomarkers. Therefore, RNA-seq is an ideal tool for defining the gene expression profiles or transcriptomes of orbital lymphoproliferative disorders.

Several previous studies used NGS to analyze IgG4-RD for the purposes of elucidating the effects of steroid treatment and discovering for biomarker [18,19], but these studies did not compare IgG4-ROD and other lymphoproliferative disorders. In this study, we investigated the expression of RNA in lesions of IgG4-ROD in biopsy specimens compared with various controls comprising adipose tissues adjacent to the IgG4-ROD lesion, orbital MALT lymphoma, RLH, and normal lacrimal gland tissues to search for new biomarkers and elucidate the pathophysiology of IgG4-ROD.

## 2. Materials and Methods

### 2.1. Patients

This study included 71 patients with lymphoproliferative disorders who visited Tokyo Medical University Hospital of Ophthalmology between 2016 and 2020.

RNA-seq was conducted using biopsy specimens containing typical lesions obtained from IgG4-ROD patients (*n* = 3). As controls, biopsy specimens of adjacent adipose tissue from the same IgG4-ROD patients, orbital MALT lymphoma possibly derived from the lacrimal gland, RLH, and normal lacrimal gland tissues were used (*n* = 3 each). The normal lacrimal gland tissues were collected to examine them for infiltration at the time of removing other orbital tumor (schwannoma, 1 case) or for the purpose of diagnosing lacrimal gland enlargement (lacrimal gland dislocations, 2 cases). All biopsy specimens were stored frozen within 30 minutes after collection and each sample was stored at −80 °C until analysis. The demographic and laboratory features of the subjects and controls studied by RNA-seq are summarized in Table 1. The mean (±SD) serum IgG4 level was higher in the IgG4-ROD patients than in the patients with MALT lymphoma and RLH (1253.3 ± 770.3, 22.7 ± 10.1 and 53.1 ± 53.1 mg/dL, respectively; not measured in lacrimal gland cases).

Real-time PCR was performed using biopsy specimens obtained from 30 patients with IgG4-ROD and 30 patients with orbital MALT lymphoma. The demographic and laboratory features of the study population in real-time PCR study are summarized in Table 2. The 30 samples of MALT lymphoma included the one sample used for RNA-seq, but the 30 samples of IgG4-ROD did not include those used in RNA-seq.

The diagnoses of IgG4-ROD, orbital MALT lymphoma, and RLH were based on clinical, radiographic, histologic and flow cytometric studies, and molecular genetic analyses, such as gene rearrangement, in the biopsies. In particular, the diagnosis of IgG4-ROD was made in accordance with the published criteria [6]. Briefly, the criteria were based on three characteristic findings: (1) enlargement of the orbital tissues, marked diffuse lymphoplasmacytic infiltrate with either fibrosis or sclerosis, (2) tissue IgG4/IgG ratio > 40% and tissue IgG4^+^ plasma cells >50 cells/high power field, and (3) serum IgG4 level above 135 mg/dL. Definite IgG4-ROD was diagnosed when (1), (2) and (3) were present. Probable IgG4-ROD was diagnosed when (1) and (2) were present. Twenty-eight patients had definitive and five had probable IgG4-ROD. 

Written informed consent was obtained from all the participants of the study. The study was also approved by the Ethics Committee of the Tokyo Medical University Hospital, Tokyo, Japan (number: SH3281). All investigations were conducted according to the principles of the Helsinki declaration.

### 2.2. RNA-Seq Analysis for Differentially Expressed Genes and Pathway Analysis

Total RNA was extracted from each tissue using TRIZOL (Thermo Fisher Scientific, Waltham, MA, United States of America (USA)), and RNA integrity was measured by a Bioanalyzer RNA 6000 Pico Kit (Agilent Technologies, Santa Clara, CA, USA). The depletion of ribosomal RNA (rRNA) and RNA-seq library preparation was performed with a NEBNext rRNA Depletion Kit and a NEBNext Ultra Directional RNA Library Prep Kit, respectively (New England Biolabs, Ipswich, MA, USA). Sequencing for 2 × 36-bp paired-end reads was performed with NextSeq500 (Illumina, San Diego, CA, USA). FASTQ files were imported to the CLC Genomics Workbench (Qiagen, v10.1.1, Hilden, Germany). Reads were mapped to the hg19 human reference genome and quantified against 57,773 genes. Differentially expressed genes (DEGs) in IgG4-ROD were defined as *p* < 0.05 and absolute FC (|FC|) ≥ 2 and were calculated using Empirical Analysis with the DEG tool.

Pathways associated with up-regulated DEGs were analyzed using Reactome [20]. Up to 3000 genes can be input into Reactome. When there were 3001 or more DEGs, the top 3000 genes according to their *p* values were analyzed. The Reactome provides many cellular processes as a network of molecular transformations under a single consistent data model [21].

### 2.3. Real-Time Quantitative PCR

The total RNA was extracted using a miRNAeasy Mini Kit (Qiagen GmbH, Hilden, Germany) and reverse transcribed using a High Capacity RNA-to-cDNA kit (Thermo Fisher Scientific) according to the manufacturer’s instructions. Matrix metallopeptidase 12 (MMP12), secreted phosphoprotein 1 (SPP1), and *β*-actin were detected using TaqMan probes (Assay ID: Hs00159178_m1, Hs00959010_m1, and Hs99999903_m1, respectively; Thermo Fisher Scientific) with a 7900 HT Fast Real-Time PCR system (Thermo Fisher Scientific). The PCR cycling conditions were 95 °C for 10 min, followed by 40 cycles of 15 s at 95 °C and 60 s at 60 °C. Data were normalized to the expression of the *β*-actin. Relative quantification was performed using a calibration curve with placental cDNA.

### 2.4. Immunostaining

Immunostaining by mouse anti-human MMP12 (1:50, MAB919, R&D Systems, Minneapolis, MN, USA) and rabbit anti-human SPP1 (1:1000, HPA027541, Atlas Antibodies, Bromma, Sweden) was performed using paraffin-embedded biopsy tissues. The intensity of immunostaining and the distribution of immunoreactivity were examined.

### 2.5. Statistical Analysis

Real-time PCR data were analyzed with SPSS statistical software version 26. The statistical significance of difference was determined using a Mann–Whitney U test. A *p* value less than 0.05 was considered statistically significant. Principal component analysis (PCA) was performed using R (3.6.2.) [22]. PCA was used to discriminate the different biological samples based on the distances of a reduced set of new variables (principal components). PCA was performed using the normalized expression levels of 8856 RNAs obtained from RNA-seq for each sample. Analysis of variance (ANOVA) was also performed, and RNAs with *p* < 0.05 were extracted. The results of the first and second principal components were used for plotting the results in two dimensions. Unsupervised hierarchical clustering analysis using DEGs was performed using an algorithm based on Pearson correlation and the average-linkage method.

## 3. Results

### 3.1. DEGs of IgG4-ROD and Other Lymphoproliferative Disorders Using RNA-Seq

RNA-seq detected 35 up-regulated DEGs and nine down-regulated DEGs in IgG4-ROD when compared with all the controls (Figure 1). An unsupervised hierarchical cluster analysis was then performed on these 44 genes to investigate variations in the DEGs between IgG4-ROD and the other controls (Figure 1, Table 3). These genes were segregated clearly into low and high expression groups for IgG4-ROD specimens but showed widely variable distributions with no consistent patterns in all the control tissues, indicating that the cluster analysis using DEGs separated IgG4-ROD from other diseases and normal tissues. These results indicate a possibility that these genes may contain diagnostic and/or therapeutic biomarkers or important factors contributing to the pathological condition of IgG4-ROD.

When compared to adjacent adipose tissue, 2698 DEGs were up-regulated and 3057 were down-regulated in IgG4-ROD. Pathway analysis using the up-regulated genes revealed that 59 pathways (including duplications) were related to IgG4-ROD (Figure 2, Table 4). The predominant pathways were related to the immune system, such as the “endosomal/vacuolar pathway”, “antigen presentation: folding, assembly, and peptide loading of class I MHC”, “immunoregulatory interactions between a lymphoid and a non-lymphoid cell”, “interferon alpha/beta signaling”, and “ER–phagosome pathway”.

Compared with orbital MALT lymphoma, 1718 DEGs were up-regulated and 705 were down-regulated in IgG4-ROD. Pathway analysis using the up-regulated genes revealed that 64 pathways were related to IgG4-ROD (Figure 3, Table 5). The predominant pathways were related to the immune system, such as “classical antibody-mediated complement activation”, “FCGR activation”, “regulation of complement cascade”, “CD22 mediated BCR regulation”, and “complement cascade”.

When compared with RLH, 439 DEGs were up-regulated and 342 were down-regulated in IgG4-ROD. Pathway analysis using the up-regulated genes revealed that 43 pathways (including duplications) were related to IgG4-ROD (Figure 4, Table 6). The predominant pathways were related to the immune system, such as “classical antibody-mediated complement activation”, “FCGR activation”, “CD22 mediated BCR regulation”, “regulation of actin dynamics for phagocytic cup formation”, and “role of phospholipids in phagocytosis”. IgG4-ROD and RLH were distributed close to each other in the PCA plot (Figure 5), suggesting a possibility that the two diseases share similarities.

In total, 2897 up-regulated and 3554 down-regulated DEGs were detected in RLH compared with normal lacrimal gland tissue; among them, 2262 and 2875 DEGs, respectively, were also detected in IgG4-ROD. Appendix A lists the down-regulated and up-regulated DEGs showing low *p* values in RLH compared with the lacrimal gland; all these genes showed similar dysregulated patterns in IgG4-ROD. However, RNA-seq detected 781 DEGs in IgG4-ROD compared with RLH, which could be used for further analysis. The normalized mRNA expression levels in IgG4-ROD and the other lymphoproliferative disorders obtained from RNA-seq were subject to PCA, and the plot of the first principal components versus that of the second principal components is shown in Figure 5. IgG4-ROD, MALT lymphoma, and RLH were distributed relatively close to each other, while adipose tissue and lacrimal gland tissue were distributed in distinctly different quadrants. However, it is possible to explicitly divide these groups into five groups (IgG4-ROD, adipose tissue in IgG4-ROD, MALT lymphoma, RLH, and normal lacrimal gland tissue) by the first component and second component. When compared with normal lacrimal gland tissue, 3026 DEGs were up-regulated, and 3513 were down-regulated, in IgG4-ROD. Pathway analysis using the up-regulated genes revealed that 188 pathways (including duplications) are likely related to IgG4-ROD (Figure 6, Table 7). The predominant pathways were related to the cell cycle, such as “cell cycle”, “cell cycle, mitotic”, and “cell cycle checkpoints”, and to the immune system, such as “cytokine signaling in immune system” and “generation of second messenger molecules”.

An unsupervised hierarchical cluster analysis was then performed to investigate the variations in DEGs between IgG4-ROD and each control (Figure 2, Figure 3, Figure 4 and Figure 6b). In all the cluster analyses, IgG4-ROD was clearly separated from each control.

Pathway analysis identified many pathways related to the immune system (Table 8), suggesting that immune reaction is strongly involved in the pathology of IgG4-ROD compared with other diseases and non-disease controls. The complete results of all pathway analyses are shown in Appendix A.

### 3.2. Validation of DEGs by Real-Time Quantitative PCR

To validate the DEGs identified by RNA-seq, we performed real-time PCR using biopsy specimens (IgG4-ROD, *n* = 30; orbital MALT lymphoma, *n* = 30). MMP12 and SPP1, two genes that showed low *p* values when compared to any of the control tissues in RNA-seq (*p* = 1.51×10^−5^and 1.12 × 10^−7^, respectively, versus adipose tissue; 3.06 × 10^−9^ and 9.74 × 10^−9^ versus orbit MALT lymphoma; 1.92 × 10^−5^ and 4.32 × 10^−5^ versus RLH; and 2.25 × 10^−11^ and 3.30 × 10^−8^ versus lacrimal gland) (Figure 2, Figure 3, Figure 4 and Figure 6) were selected for validation. Real-time PCR showed that the expression levels of MMP12 and SPP1 in the biopsy specimens of IgG4-ROD were significantly higher than those of orbital MALT lymphoma (*p* = 2.7141 × 10^−8^ and 1.2044 × 10^−7^, respectively) (Figure 7). There was no sex difference in the expression levels of MMP12 and SPP1 in both diseases.

### 3.3. Immunostaining

Representative micrographs of immunostaining for MMP12 and SPP1 in IgG4-ROD, MALT lymphoma, RLH, and lacrimal gland tissues are shown in Figure 8. In the IgG4-ROD tissue, many MMP12-positive cells were present not only in lymphoid follicles but also in fibrotic areas, and SPP1 was intensely immunostained in the fibrotic part. On the other hand, MMP12 and SPPI reactivities were weak or absent in the other three tissue samples. These findings suggest that MMP12 is involved in fibrosis and follicle formation, while SPP1 is involved in fibrosis.

Immunostaining indicated expression of MMP12 and SPP1 at the protein level in IgG4-ROD biopsy tissues, suggesting that these molecules play a role in pathogenesis and could thus serve as new biomarkers for IgG4-ROD.

## 4. Discussion

In this study, we performed a comprehensive RNA analysis of lacrimal gland-derived IgG4-ROD using adipose tissue adjacent to the IgG4-ROD lesion, orbital MALT lymphoma, RLH, and normal lacrimal gland tissue as controls. Orbital MALT lymphoma and RLH were disease controls, while adipose tissue from the same patients with IgG4-ROD controlled for individual differences in gene expression, and normal lacrimal gland tissue controlled for tissue-specific gene expression. The results of the RNA-seq and pathway analysis suggest that IgG4-ROD is strongly related to the immune system and the up-regulation of MMP12 and SPP1. 

The DEGs and pathways detected by RNA-seq for IgG4-ROD versus the lacrimal gland may provide insight into the pathology of IgG4-ROD. Fibrosis and class switches to IgG4 are characteristics of the pathology of IgG4-RD. Fibrosis in IgG4-RD is considered to be caused by transforming growth factor-*β* (TGF-*β*) secreted by Tregs and CD4 + cytotoxic T cells (CD4 + CTLs) [3]. CD68 + CD163 + alternatively activated (M2) macrophages are abundantly present in IgG4-RD tissues and are involved in the orchestrated immune reaction by regulating cytokine production and fibrosis [23,24,25].

Moreover, binding of the dendritic cells and T cells to B cells via BAFF and CD40 together with the activation of AID by various cytokines has been reported to induce class switching [26]. Furthermore, class switching to IgG4 may be induced by the interleukin (IL)-21 secreted by Tfh in IgG4-RD [27]. Compared to lacrimal tissue, not only TGFB1 but also genes related to the surface markers of cells secreting TGF-*β* (CD4 + CTL marker, CD4; Treg markers, CD25 and FOXP3), as well as multiple genes related to class switching (BAFF, BAFFR, CD40, CD40L, AID, IL21) and Tfh (ICOS, CXCR5), were up-regulated in IgG4-ROD. Thus, the DEGs detected by RNA-seq demonstrate the existence of multiple genes that are implicated in the pathology of IgG4-RD in previous reports, as well as novel DEGs that should be further investigated.

In the pathway analysis using DEGs detected in IgG4-ROD versus orbital MALT lymphoma, the pathways related to the complement system dominated. Some patients with IgG4-RD have hypocomplementemia [28], suggesting the occurrence of antigen–antibody reactions in the body. Since Chlamydia psittaci DNA has been found in MALT lymphoma samples [29], antigen–antibody reactions to infection may also take place in MALT lymphoma. However, infection only serves as a trigger, and the leading cause of the pathology seems to be the self-proliferation of tumor cells due to MALT1 translocation under transcriptional control in the IgH enhancer region [30,31], activated by infection. In IgG4-ROD, however, the complement pathway is activated, and the complement is depleted. Thus, the leading cause of the pathology is likely antigen–antibody reactions to infections [4]. Since MALT lymphoma is speculated to arise from a background of IgG4-RD [11,12], both diseases may have infection as a common background.

Notably, IgG4 is regarded as an anti-inflammatory antibody because of its poor capacity to bind to Fc receptors [32], Therefore, one possible mechanism for activation of the complete pathway is that other IgG subclasses, such as IgG1 and IgG2a, bind to Fc receptors. A similar mechanism was proposed to explain why hypocomplementemia occurs frequently in IgG4-ROD patients despite the relative inability of IgG4 to fix the complement [33]. In this study, FCGR1A was up-regulated in IgG4-ROD. This gene encodes a receptor with high affinity for the Fc region of IgG, suggesting that the complement’s pathway may be activated by binding to a subclass other than IgG4.

RLH is part of the spectrum of orbital lymphoproliferative disorders [34] and is considered to be a consequence of the chronic inflammatory response of lymphoid cells to irritating or antigenic stimuli [35,36]. Differentiation between IgG4-ROD and RLH is often difficult in the results of histology, gene rearrangement, and flow cytometry [5]. Indeed, fewer DEGs were detected in IgG4-ROD compared to RLH than with the other control tissues. Furthermore, PCA using the normalized expression levels of the 8856 RNAs obtained from RNA-seq showed a close distribution of IgG4-ROD and RLH in terms of their genetic features, which may reflect the similar clinical phenotypes of the two conditions. This finding suggests that the altered genes are almost the same in both diseases. Indeed, some genes related to autoimmune diseases, such as BACH2, C17orf99, CCL19, and IL6R, showed high expression in IgG4-ROD when compared to tissues other than RLH but no difference when compared to RLH, suggesting that the two diseases may involve an autoimmune etiology [35]. If this hypothesis holds true, it is not surprising that there is considerable overlap in the gene alterations between the two conditions. An interesting question is how IgG4-ROD differs from RLH at the molecular level. To the best of our knowledge, there is no report comparing RLH with IgG4-ROD. In this study, we found for the first time that RNA-seq was able to identify 781 DEGs for IgG4-ROD versus RLH. Until now, IgG4 was the only factor distinguishing between the two, and it was difficult to search for other biomarkers. RNA-seq detected 781 DEGs, indicating the possibility of searching for new biomarkers. When we performed a preliminary pathway analysis using genes that were up-regulated in RLH relative to IgG4-ROD, pathways including “collagen degradation” were identified (Appendix A). In addition, the increased expression of BMP2 with anti-fibrotic action in RLH has been reported [37]. These findings suggest that anti-fibrosis may be promoted in RLH (or attenuated in IgG-ROD) with differential biomarkers between IgG4-ROD and RLH.

When common DEGs were extracted while excluding adjacent adipose tissue, 96 genes remained as DEGs. The genes obtained from this analysis are putatively involved in diseases involving IgG4 and IgE. However, there was no difference in the expression of these genes in comparison with the adjacent adipose tissue. This suggests that the surrounding tissues may also be affected by IgG4-ROD. However, the adipose tissue was a control tissue obtained from the same individuals, so the possibility of individual differences may be eliminated. Since there are no reports featuring analyses using different tissue samples collected from the same patient, the present findings could offer valuable data. In addition, although tissues from the lacrimal gland were also used in this study as a control, IgG4-ROD also occurs in tissues other than the lacrimal gland [6]. The presence of lymphocyte clusters in adipose tissue has been reported [38] and may be responsible for lymphoproliferative disorders. Based on the above, adipose tissue from the same patient with IgG4-ROD remains an important control, but the possibility that this tissue is affected by the lesion should be fully addressed.

This study identified two novel genes involved in IgG4-ROD: MMP12 and SPP1. Up-regulation of these genes in IgG4-ROD relative to orbital MALT lymphoma was validated by real-time PCR, and increased expression at the protein level in the IgG4-ROD lesion compared to the control tissues was verified by immunostaining. SPP1 encodes osteopontin, which is involved in the attachment of osteoclasts to the mineralized bone matrix. MMP12 encodes an enzyme that decomposes the extracellular matrix. SPP1 was reported to be involved in fibrosis in various diseases [39,40,41,42]. The histopathology of IgG4-RD is characterized by fibrosis and lymphoid follicle formation. Until now, the TGF-β secreted by CTLs and Tregs was implicated in fibrosis [3], but it is possible that the presence of SPP1 strongly activates TGF-*β*-induced fibrosis. This is inferred from our finding that SPP1 is strongly expressed in the fibrotic part of IgG4-ROD tissue.

In this study, commonly up-regulated DEGs among IgG4-ROD and control tissues, MMP12 showed low *p* values when IgG4-ROD was compared to each control tissue. It was also reported that M2 macrophages secrete MMP12, which is an elastase, and are involved in the pathological conditions of contact dermatitis [43]. Generally, M2 macrophages have an anti-inflammatory effect [44,45,46] and are related to IgG4-RD [47]. The exhibition of an anti-inflammatory effect suggests the presence of inflammation. Some IgG4-ROD patients resist systemic steroid treatment. If IgG4-ROD is indeed an immune reaction to a pathogen, the disease would recur as the effect of steroids gradually decreases, resulting in failure to suppress the immune reaction. In the pathway analysis, pathways related to the immune system were extracted when IgG4-ROD was compared with any control tissue, strongly suggesting the possibility that immunity is involved in the pathological condition. We hypothesize that in IgG4-ROD, an infection triggers an inflammatory reaction in the lacrimal gland, and M2 macrophages secrete elastase with an anti-inflammatory effect. MMP12 is included in the pathways involved in the extracellular matrix and collagen degradation, although there was no significant difference in these pathways observed in this study. MMP12 promotes fibrosis in *Schistosoma mansoni* infection [48] and may also be involved in fibrosis in IgG4-ROD.

In our experiment, in which we identified the biopsy gene profile of RLH, we found that the gene profile of RLH was remarkably similar to that of IgG4-ROD and was characterized by decreased expression of MMP12 and SPP1 compared to IgG4-ROD (Appendix A). Notably, we were able to differentiate these two very similar diseases at the RNA expression level. Pathologically, IgG4-ROD and RLH are distinctly different in terms of the presence or absence of fibrosis; at the gene level, IgG4-ROD differs from RLH in not only showing the up-regulation of genes related to fibrosis, such as SPP1 and MMP12, but also the attenuation of anti-fibrotic elements, such as collagen degradation, that may lead to fibrosis.

Limitations of this study include its retrospective design and relatively small number of cases collected from a single institution, which might have produced a selection bias and confounding bias. Due to the lack of comparable studies on IgG4-ROD including IgG4-RD with other diseases, it is not possible to compare the present findings with those of other studies. A prospective study with a large number of clinical samples from multiple centers is required to validate the present findings and further investigate other clinical manifestations of IgG4-ROD and other orbital tumors to diagnose clinical subtypes and elucidate their pathogeneses. In this study, we used real-time PCR to verify only two genes with enhanced expression in RNA-seq. However, it is necessary to carry out additional studies by quantifying other genes and analyzing their protein expression levels. Furthermore, MMP12 and SPP1 should be analyzed in larger samples of IgG4-ROD and other IgG4-RDs, and it is also necessary to examine the dynamics of these genes and proteins in IgG4-positive MALT lymphoma.

## 5. Conclusions

To the best of our knowledge, this is the first report of a comprehensive mRNA analysis of IgG4-ROD using RNA-seq. We succeeded in detecting not only previously reported genes but also novel genes that that may serve as biomarkers or may be involved in the pathogenesis of IgG4-ROD. In addition, these results may provide novel insights for finding new therapeutic target molecules.

## Figures and Tables

**Figure 1 jcm-09-03458-f001:**
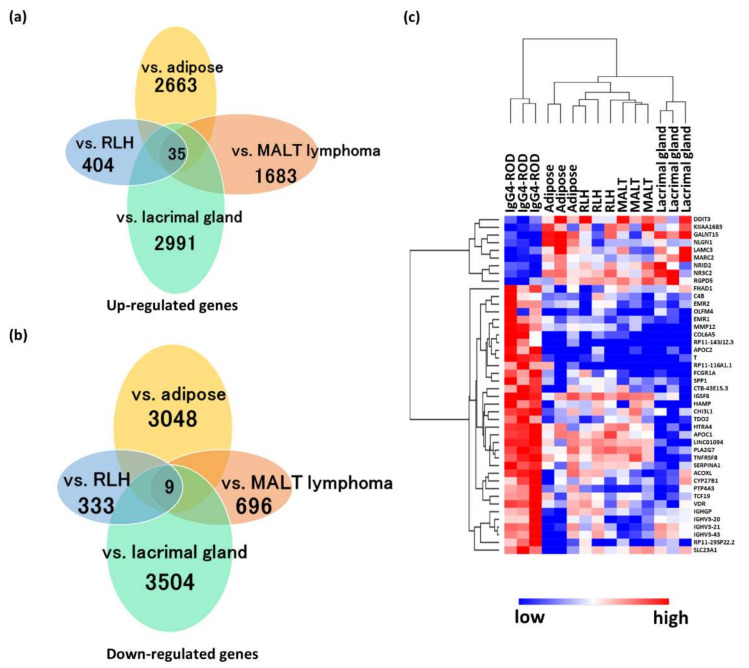
Common differentially expressed genes (DEGs) in immunoglobulin G4-related ophthalmic disease compared (IgG4-ROD) with the control tissues. (**a**,**b**) Venn diagrams show the numbers of messenger RNAs up-regulated (**a**) or down-regulated (**b**) compared to each control. (**c**) Heatmap obtained from unsupervised hierarchical clustering analysis using common DEGs. The red to blue spectrum indicates high to low values. MALT: Mucosa-associated lymph tissue, RLH: Reactive lymphoid hyperplasia.

**Figure 2 jcm-09-03458-f002:**
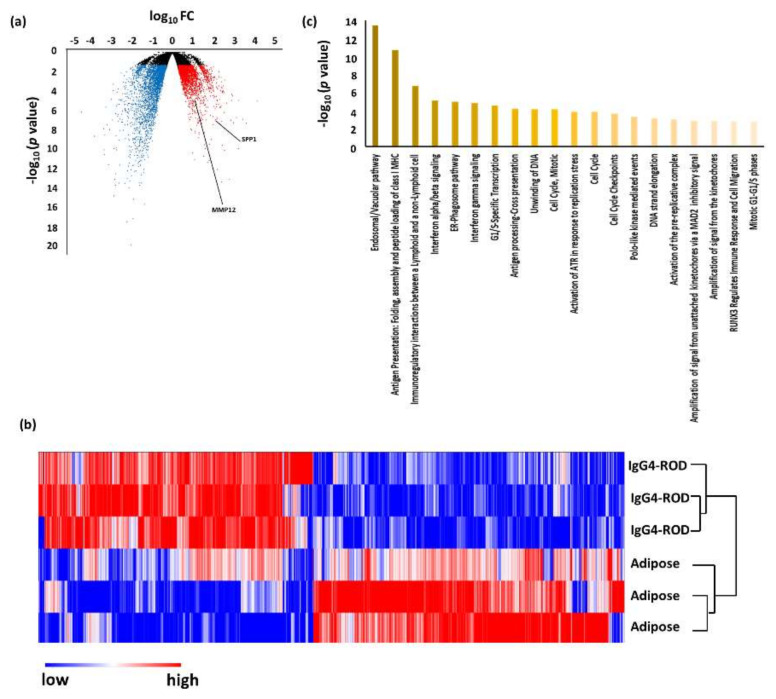
Results of RNA sequencing for the tissue of immunoglobulin G4-related ophthalmic disease (IgG4-ROD) compared with adjacent adipose tissue. (**a**) Volcano plot of messenger RNAs (mRNAs). Blue dots, mRNA down-regulation; red dots, up-regulation; black dots, nonsignificant expression. Horizontal axis, fold change (FC); vertical axis: *p* value. (**b**) Heatmap obtained from unsupervised hierarchical clustering analysis using differentially expressed genes. The red to blue spectrum corresponds to high to low values. (**c**) Histogram showing pathways enriched in IgG4-ROD. Vertical axis: −log10 (*p* value).

**Figure 3 jcm-09-03458-f003:**
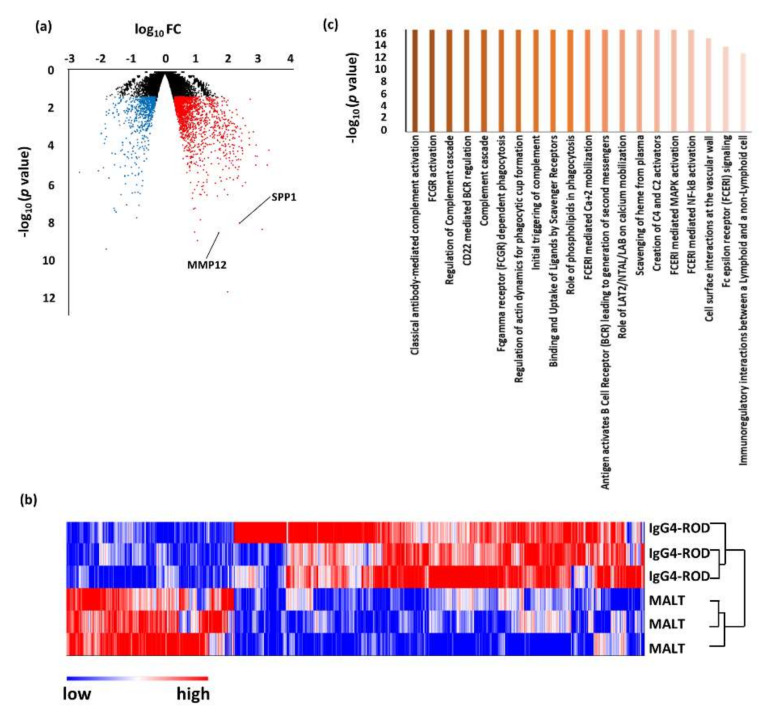
Results of RNA sequencing for tissue of immunoglobulin G4-related ophthalmic disease (IgG4-ROD) compared with orbital mucosa-associated lymphoid tissue (MALT) lymphoma. (**a**) Volcano plot of messenger RNAs (mRNAs). Blue dots, mRNA down-regulation; red dots, up-regulation; black dots, nonsignificant expression. Horizontal axis, fold change (FC); vertical axis: *p* value. (**b**) Heatmap obtained from unsupervised hierarchical clustering analysis using differentially expressed genes. The red to blue spectrum corresponds to high to low values. (**c**) Histogram showing pathways enriched in IgG4-ROD. Vertical axis: −log10 (*p* value).

**Figure 4 jcm-09-03458-f004:**
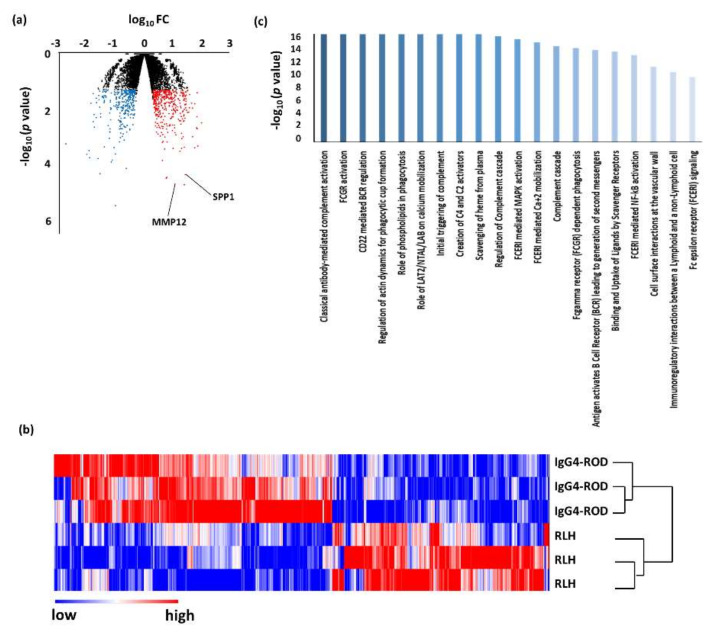
Results of RNA sequencing for the tissues of immunoglobulin G4-related ophthalmic disease (IgG4-ROD) compared with reactive lymphoid hyperplasia. (**a**) Volcano plot of messenger RNAs (mRNAs). Blue dots, mRNA down-regulation; red dots, up-regulation; black dots, nonsignificant expression. Horizontal axis, fold change (FC); vertical axis: *p* value. (**b**) Heatmap obtained from unsupervised hierarchical clustering analysis using differentially expressed genes. The red to blue spectrum corresponds to high to low values. (**c**) Histogram showing pathways enriched in IgG4-ROD. Vertical axis: −log10 (*p* value).

**Figure 5 jcm-09-03458-f005:**
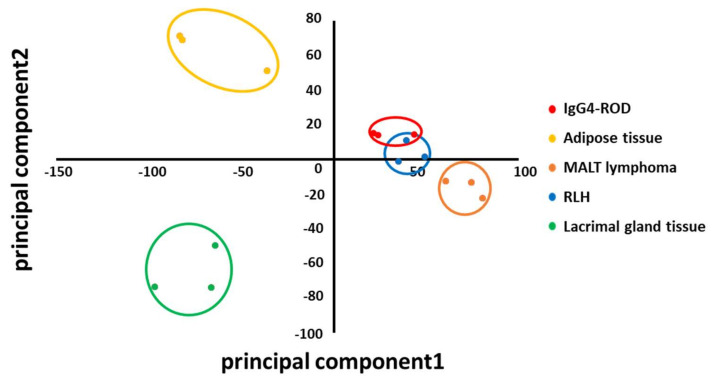
Principal component analysis plot separating immunoglobulin G4-related ophthalmic disease (IgG4-ROD), adipose tissue, reactive lymphoid hyperplasia (RLH), lacrimal gland, and mucosa-associated lymphoid tissue (MALT) lymphoma using the results of RNA sequencing. Three samples in each group were analyzed. Each dot denotes one sample, and the three samples of each group are enclosed in a circle. Red dots: IgG4-ROD, yellow dots: adjacent adipose tissue, orange dots: MALT lymphoma, blue dots: RLH, green dots: lacrimal gland.

**Figure 6 jcm-09-03458-f006:**
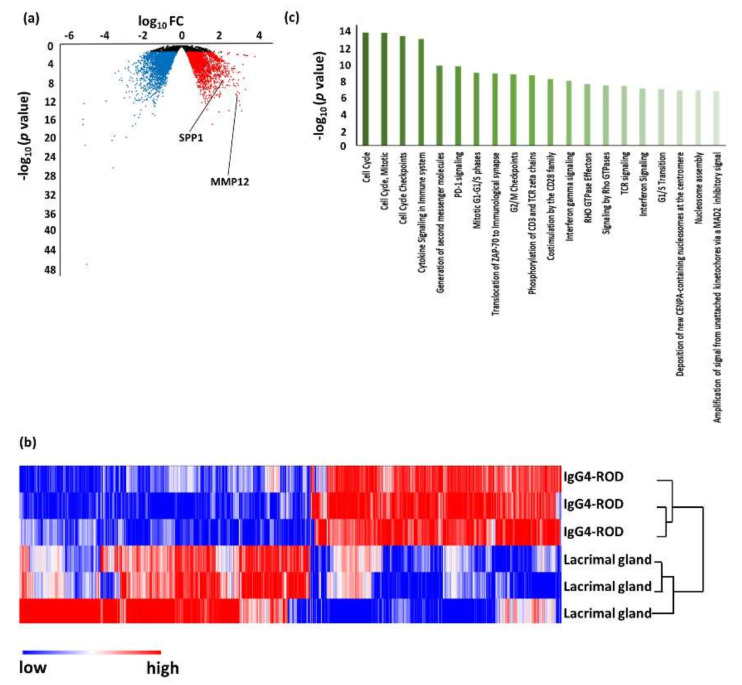
Results of RNA sequencing for the tissue of immunoglobulin G4-related ophthalmic disease (IgG4-ROD) compared with a normal lacrimal gland. (**a**) Volcano plot of messenger RNAs (mRNAs). Blue dots, mRNA down-regulation; red dots, up-regulation; black dots, nonsignificant expression. Horizontal axis, fold change (FC); vertical axis: *p* value. (**b**) Heatmap obtained from unsupervised hierarchical clustering analysis using differentially expressed genes. The red to blue spectrum corresponds to high to low values. (**c**) Histogram showing pathways enriched in IgG4-ROD. Vertical axis: −log10 (*p* value).

**Figure 7 jcm-09-03458-f007:**
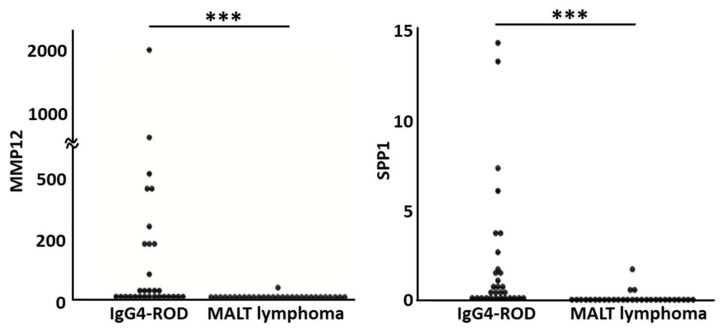
Results of real-time PCR for validation of differentially expressed genes obtained from RNA sequencing. *** Messenger RNA of MMP12 and SPP1 in immunoglobulin G4-related ophthalmic disease is significantly higher than mucosa-associated lymphoid tissue lymphoma (*p* = 2.7141 × 10^−8^ and 1.2044 × 10^−7^, respectively). Statistical analysis was performed by a Mann–Whitney *U* test (*n* = 30). IgG4-ROD: Immunoglobulin G4 related ophthalmic disease, MALT: Mucosa-associated lymph tissue, MMP12: Matrix metallopeptidase 12, SPP1: Secreted phosphoprotein 1.

**Figure 8 jcm-09-03458-f008:**
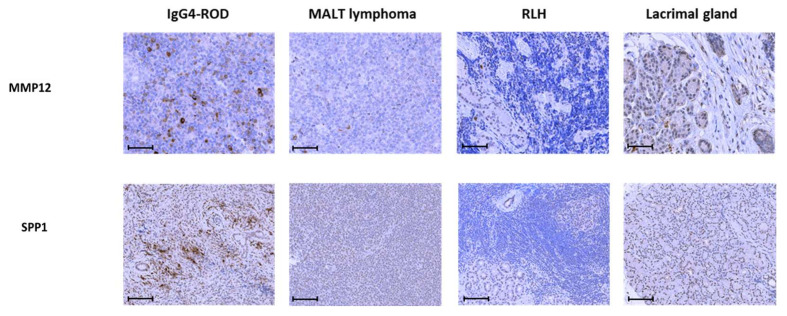
Representative immunohistochemical staining patterns of formalin-fixed, paraffin-embedded sections (MMP12: magnification 400×, bar: 50 μm, SPP1: magnification 200×, bar: 100μm). The upper row is immunoreactivity of MMP12, and the lower row is immunoreactivity of SPP1. From left column to right column, IgG4-ROD, MALT lymphoma, RLH, and the lacrimal gland tissue section are shown. Many MMP12 positive cells are observable in the fibrotic areas and follicular areas of IgG4-ROD tissues. SPP1 is intensely stained in the fibrotic part of IgG4-ROD. IgG4-ROD: Immunoglobulin G4 related ophthalmic disease, MALT: Mucosa-associated lymph tissue, RLH: Reactive lymphoid hyperplasia, MMP12: Matrix metallopeptidase 12, SPP1: Secreted phosphoprotein 1.

**Table 1 jcm-09-03458-t001:** Clinical and laboratory features of patients with IgG4-ROD and controls in the RNA sequencing study.

	IgG4-ROD	MALT Lymphoma	RLH	Lacrimal Gland
Number	3	3	3	3
Sex (male:female)	2:1	1:2	1:2	0:3
Age (year) Mean ± SD (range)	48.7 ± 19.4 (22–69)	67.8 ± 2.7 (64–70)	34.9 ± 5.7 (29–42)	32.3 ± 12.4 (17–47)
IgG4 (mg/dL) mean ± SD	1253.3 ± 770.3	22.7 ± 10.1	53.1 ± 12.9	-

IgG4-ROD: immunoglobulin G4 related ophthalmic disease, MALT: mucosa-associated lymph tissue, RLH: reactive lymphoid hyperplasia, SD: standard deviation.

**Table 2 jcm-09-03458-t002:** Clinical and laboratory features of patients with IgG4-ROD and MALT lymphoma in the real-time PCR study.

	IgG4-ROD	MALT Lymphoma
Number	30	30
Sex (male: female)	13:17	17:13
Age (year) mean ± SD (range)	55.5 ± 14.7 (27–85)	71.0 ± 10.3 (52–91)
IgG4 (mg/dL) mean ± SD	514.3 ± 403.3	107.9 ± 200.0

IgG4-ROD: immunoglobulin G4 related ophthalmic disease, MALT: mucosa-associated lymph tissue, SD: standard deviation.

**Table 3 jcm-09-03458-t003:** List of DEGs when the IgG4-ROD tissue was compared to all control tissues.

Up-Regulated DEGs	Down-Regulated DEGs
ACOXL	GALNT15
APOC1	RGPD5
APOC2	NR3C2
C4B	MARC2
CHI3L1	DDIT3
COL6A5	LAMC3
CTB-43E15.3	NR1D2
CYP27B1	NLGN1
EMR1	KIAA1683
EMR2	
FCGR1A	
FHAD1	
HAMP	
HTRA4	
IGHGP	
IGHV3-20	
IGHV3-21	
IGHV3-43	
IGSF6	
LINC01094	
MMP12	
OLFM4	
PLA2G7	
PTP4A3	
RP11-116A1.1	
RP11-143J12.3	
RP11-295P22.2	
SERPINA1	
SLC23A1	
SPP1	
T	
TCF19	
TDO2	
TNFRSF8	
VDR	

DEGs: Differentially expressed genes. IgG4-ROD: Immunoglobulin G4 related ophthalmic disease.

**Table 4 jcm-09-03458-t004:** Differentially expressed genes of IgG4-ROD versus the adjacent adipose tissue: top 40 in descending *p* value.

Gene Name	Fold Change	*p*-Value
SLIT3	−98.4337563	1.40371 × 10^−20^
GPC3	−152.1472889	3.63961 ×^ 10^−18
NTRK2	−76.43198001	7.63906 × 10^−18^
TGFBR3	−21.91554793	2.35247 × 10^−16^
TSKU	−72.31347949	7.88449 × 10^−16^
GPD1	−654.4444489	1.76224 × 10^−15^
AKR1C1	−120.6905584	1.16455 × 10^−14^
CPE	−38.72239459	2.09594 × 10^−14^
PPP1R1A	−547.0382931	2.16735 × 10^−14^
PODN	−192.7417821	3.39541 × 10^−14^
IGFBP6	−379.1743688	3.68508 × 10^−14^
PPAP2B	−30.61599516	3.73239 × 10^−14^
GSN	−39.35352686	3.87062 × 10^−14^
IL17REL	229.2877389	4.55926 × 10^−14^
OSR2	−110.7599045	5.74471 × 10^−14^
PLIN4	−592.416461	8.76055 × 10^−14^
LTF	564.0886917	8.79702 × 10^−14^
TNXB	−146.986527	1.17268 × 10^−13^
AQPEP	−90.84644713	2.03543 × 10^−13^
KANK2	−31.23331966	2.56161 × 10^−13^
ADH1B	−518.8872444	3.19671 × 10^−13^
COX7A1	−24.28261826	4.39214 × 10^−13^
SMOC1	−64.28855502	5.81357 × 10^−13^
NEGR1	−57.41166123	6.57915 × 10^−13^
CHRDL1	−141.0949914	6.808 × 10^−13^
MAOA	−98.27241905	8.53169 × 10^−13^
TNS1	−23.71338196	9.00895 × 10^−13^
NPHS1	145.7234025	9.17012 × 10^−13^
NAV3	−43.29399038	1.79482 × 10^−12^
CLEC3B	−129.0389168	2.05641 × 10^−12^
SRPX	−82.7390596	2.2354 × 10^−12^
BMP4	−35.19126943	2.44309 × 10^−12^
FBLN2	−71.14840737	2.52572 × 10^−12^
LAMA4	−34.33402456	2.58904 × 10^−12^
AGTR1	−340.16385	2.62539 × 10^−12^
CNTFR	−1164.982361	2.77156 × 10^−12^
CD34	−44.1295727	3.70286 × 10^−12^
ADIRF	−117.0194857	3.96587 × 10^−12^
KEL	49.28469675	4.6871 × 10^−12^
FZD4	−41.36375766	5.26482 × 10^−12^

IgG4-ROD: Immunoglobulin G4 related ophthalmic disease.

**Table 5 jcm-09-03458-t005:** Differentially expressed genes of IgG4-ROD versus orbital MALT lymphoma: top 40 in descending p value.

Gene Name	Fold Change	*p*-Value
PADI2	91.58382541	2.1773 × 10^−12^
FCRL4	−70.09053891	4.19504 × 10^−10^
MGLL	10.47760485	1.16413 × 10^−9^
MMP12	48.21503566	3.05605 × 10^−9^
EYA2	8.528054913	3.22527 × 10^−9^
IGHG4	1126.488944	4.65877 × 10^−9^
SPP1	215.4347982	9.74256 × 10^−9^
ALDH1L2	12.90968552	1.48731 × 10^−8^
ARHGAP24	−7.710194267	1.87049 × 10^−8^
WNT3	−34.00034629	3.78215 × 10^−8^
WSCD2	−37.78424695	4.72911 × 10^−8^
ITGA6	7.990302303	5.43839 × 10^−8^
LGMN	8.788326656	5.99957 × 10^−8^
ADAM29	−39.81529876	6.09785 × 10^−8^
OSBPL10	−6.485891987	8.63561 × 10^−8^
RP11-38J22.6	−16.17176916	9.1141 × 10^−8^
SASH1	7.157215503	1.99315 × 10^−7^
PLA2G2A	44.02915635	2.66684 × 10^−7^
CA2	20.70081256	3.06011 × 10^−7^
GAS6	13.57656547	3.10603 × 10^−7^
SOX5	−23.6043411	3.13093 × 10^−7^
IGHV3-21	92.96849744	3.20167 × 10^−7^
HID1	12.26150423	3.28431 × 10^−7^
CYB5R2	−6.002340514	3.49966 × 10^−7^
SLC30A4	7.403137108	3.83488 × 10^−7^
ADAMTS6	−10.67948276	3.8833× 10^−7^
CSGALNACT1	−6.942823556	4.11803 × 10^−7^
UCHL1	17.37051895	4.16921 × 10^−7^
CNR1	−8.606871826	4.25411 × 10^−7^
KIAA1244	73.76997554	5.78475 × 10^−7^
ATP10A	8.654788169	5.79467 × 10^−7^
EMR2	15.11956486	5.81809 × 10^−7^
C1QA	14.14827005	8.39642 × 10^−7^
IGKV2-28	271.6998027	9.1622 × 10^−7^
THNSL2	15.65957129	9.22407 × 10^−7^
CCDC50	−5.149996654	9.42061 × 10^−7^
KCNJ5	10.04335239	9.63907 × 10^−7^
C10orf128	−7.729250078	1.03206 × 10^−6^
DGKG	−17.08198927	1.16567 × 10^−6^
MLPH	38.01969836	1.2827 ×10^−6^

IgG4-ROD: Immunoglobulin G4 related ophthalmic disease, MALT: mucosa-associated lymph tissue.

**Table 6 jcm-09-03458-t006:** Differentially expressed genes of IgG4-ROD versus RLH: top 40 in descending *p* value.

Gene Name	Fold Change	*p*-Value
FAM177B	−10.3900113	3.25336 × 10^−6^
EMR1	25.57958237	1.86062 × 10^−5^
MMP12	11.8337076	1.92343 × 10^−5^
ALDH1L2	5.814907922	3.14093 × 10^−5^
PTP4A3	6.113035976	3.38518 × 10^−5^
SPP1	27.63935883	4.3162 × 10^−5^
AC017116.11	−36.28315709	4.64012 × 10^−5^
NT5E	-4.273902544	7.11085 × 10^−5^
CNN1	−14.56659651	0.000122289
WIPI1	4.284464213	0.000138929
ADH1B	−13.41953603	0.000143646
LCN1	−42.29922675	0.000161857
AC092299.7	−84.72321323	0.000173917
AC104057.1	−106.3580812	0.000235194
IGFBP6	−10.39244614	0.000237936
CYP21A2	18.14213394	0.000298487
CA3	−88.9511056	0.000319793
HID1	5.211793666	0.000356192
HSPA1B	10.28718484	0.000381545
CRYBB1	20.32925057	0.000384202
SIM1	59.69090341	0.000397879
PI16	−11.52129282	0.000406239
SRXN1	25.64820111	0.000415758
GALNT15	−25.66256612	0.000427794
ACTN3	76.63388014	0.000469447
ADIPOQ	−582.3821654	0.000552215
FKBP2	4.340734759	0.000613148
RP11-661A12.5	−7.507751627	0.000654035
CA2	6.297906839	0.000654227
KCNJ5	4.506254821	0.000685835
KHDRBS2	−5.213708869	0.00075486
VMO1	21.13082201	0.000854377
BFSP1	−17.07267846	0.000893836
PDK4	−7.042903709	0.000927175
TNFRSF8	5.357398909	0.000992411
HSPE1-MOB4	49.21365017	0.001001187
IGKV2-28	23.01354236	0.001013264
TNXB	−6.033905531	0.001024039
FNDC3B	3.585712045	0.001059042
APOC2	12.84388683	0.00117241

IgG4-ROD: Immunoglobulin G4 related ophthalmic disease, RLH: reactive lymphoid hyperplasia.

**Table 7 jcm-09-03458-t007:** Differentially expressed genes of IgG4-ROD versus normal lacrimal gland: top 40 in descending *p* value.

Gene Name	Fold Change	*p*-Value
CST4	−104905.8219	6.18836 × 10^−48^
PRR4	−4228.646736	4.65832 × 10^−27^
LACRT	−128185.3334	3.82972 × 10^−22^
HMGCS2	−4704.711396	3.88083 × 10^−21^
LPO	−858.7521916	1.58857 × 10^−20^
DMBT1	−3591.244728	3.18218 × 10^−20^
FAM83A	−394.4256361	5.2413 × 10^−18^
SCGB1D1	−162115.4934	1.02646 × 10^−17^
SLC2A5	43.51931001	1.09134 × 10^−17^
ATP2B2	−275.0166884	1.4644 × 10^−17^
LCN1	−167236.6822	1.55603 × 10^−16^
NRG3	−187.161496	3.32815 × 10^−16^
CHRM1	−309.6706246	1.03663 × 10^−15^
LMO4	−25.49885769	1.03941 × 10^−15^
LYZ	−158.3609865	1.98988 × 10^−15^
EDA	−34.06201821	2.07347 × 10^−15^
CCR7	97.43845229	5.62697 × 10^−15^
CNR2	1558.530919	8.97198 × 10^−15^
ALPK2	339.9116591	1.02616 × 10^−14^
AZGP1P1	−145.3980047	1.52873 × 10^−14^
PPP1R16B	22.84910658	1.73181 × 10^−14^
CGNL1	−26.19823156	8.00626 × 10^−14^
CHST1	−79.16211143	9.80791 × 10^−14^
MYH11	−28.47335481	1.4437 × 10^−13^
NPHS1	1208.116136	1.69991 × 10^−13^
LONRF2	−22.74438049	3.35028 × 10^−13^
RASSF2	19.17351295	3.43781 × 10^−13^
SCGB2A1	−148760.8536	3.4533 × 10^−13^
TBC1D30	−29.35606139	3.88755 × 10^−13^
PPP1R1A	−335.7595576	5.30113 × 10^−13^
PLEKHG1	12.22377766	5.58532 × 10^−13^
PGR	−26.77020686	6.9827 × 10^−13^
CHRM3	−116.642127	7.47294 × 10^−13^
SLC12A2	−27.4397416	8.07107 × 10^−13^
LINC00649	25.06698425	1.06345 × 10^−12^
C22orf34	17.85464648	1.11473 × 10^−12^
DLK1	−9595.571191	1.20025 × 10^−12^
HSPG2	−31.04452153	1.28414 × 10^−12^
TNFRSF8	1123.331724	1.30319 × 10^−12^
CAMK2N1	−33.81164218	1.40062 × 10^−12^

IgG4-ROD: Immunoglobulin G4 related ophthalmic disease.

**Table 8 jcm-09-03458-t008:** Number of related pathways when the IgG4-ROD tissue was compared to each control tissue.

Pathway Type	Vs. Adjacent Adipose	Vs. MALT Lymphoma	Vs. RLH	Vs. Lacrimal Gland
Cell cycle	26	5		63
Cellular responses to external stimuli		3	5	5
Chromatin organization		2		1
Developmental biology			2	2
Disease		2		4
DNA repair	8	4		27
DNA replication	4			18
Extracellular matrix organization		1	1	
Gene expression	4	11	1	26
Hemostasis	1	2	2	1
Immune system	16	20	21	38
Metabolism of proteins		4	6	2
Metabolism of RNA		1		
Programmed cell death	1			2
Reproduction				4
Signal transduction	2	4		9
Vesicle-mediated transport		5	4	
Organelle biogenesis and maintenance			2	

MALT: Mucosa-associated lymph tissue, RLH: Reactive lymphoid hyperplasia.

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
