# Peer review of "Comprehensive Gene Analysis of IgG4-Related Ophthalmic Disease Using RNA Sequencing"

_jcm, 2020, doi:10.3390/jcm9113458_

Round 1

Reviewer 1 Report

The authors have addressed all of the concerns raised by the reviewers. 

Author Response

The authors have addressed all of the concerns raised by the reviewers. 

Response

We thank the reviewer for the thoughtful and encouraging comment.

Reviewer 2 Report

In this revision, the authors have made improvements and clarifications to the manuscript, which addressed all my concerns.

Author Response

In this revision, the authors have made improvements and clarifications to the manuscript, which addressed all my concerns.

Response

We thank the reviewer for the thoughtful and encouraging comment.

Reviewer 3 Report

Congratulations to the authors, since they have improved the paper, providing additional Tables as supplementary data 

Author Response

Congratulations to the authors, since they have improved the paper, providing additional Tables as supplementary data 

Response

We thank the reviewer for the thoughtful and encouraging comment.

This manuscript is a resubmission of an earlier submission. The following is a list of the peer review reports and author responses from that submission.

Round 1

Reviewer 1 Report

This is a very interesting and well designed study, limited only by its retrospective nature and small number of subjects from a single institution (as the authors rightfully point out in their limitations). The study presents a very promising avenue towards studying the pathology of orbital inflammatory disorders, one which may ultimately shed light on the pathogenesis of IgG4-related disease of the orbit and related conditions.

Author Response

This is a very interesting and well designed study, limited only by its retrospective nature and small number of subjects from a single institution (as the authors rightfully point out in their limitations). The study presents a very promising avenue towards studying the pathology of orbital inflammatory disorders, one which may ultimately shed light on the pathogenesis of IgG4-related disease of the orbit and related conditions.

Response

We thank the reviewer for the thoughtful and encouraging comments.

Reviewer 2 Report

The paper titled “Comprehensive gene analysis of IgG4-related ophthalmic disease using RNA sequencing”, reported on an interesting topic regarding RNA expression levels found by RNA-Seq in biopsy specimens of 3 patients diagnosed with IgG4-ROD compared to controls presenting with Mucosa-associated lymphoid tissue (MALT) lymphoma, reactive lymphoid hyperplasia (RLH), normal lacrimal gland tissue, and adjacent adipose tissue.

RNA-seq identified 35 upregulated genes including MMP12 and SPP1 in IgG4-ROD tissues when compared with all the controls, confirmed by real-time PCR and immunohistochemistry, that the authors considered having a role in the pathogenesis of Ig4-ROD and they could be considered as biomarkers.

The text and contents are understandable. However, there are some lack of information that should be added. Minor revision should be considered.

Minor English revision should be considered.

There are only few specific concerns:

The title should be changed as this topic regarding only few patients: I suggest modifying as follow: Comprehensive gene analysis of IgG4-related ophthalmic disease using RNA sequencing: a preliminary case series study.  

The hypothesis of the possible role of a triggering infection as Epstein Barr infection should be well reported and the data of the possibile nifection including serology for each patient should be shown. 

Lines 88 and 89 page 2: please add Standard deviation (SD)

Table 2 page 3: there is a mistake on the second line regarding Aex (male/female): IgG4-ROD is 13/30 or 13/17 and in MLAT lymphoma I 17/30 or 17/13?: please clarify

Line 335 page 18: please delete the point before reference number 32

Lines 352-354 page 18: please delete the sentence When… as the data is not shown by the authors or the authors should decide to show the data.

Lines 385 -387 page 19: the same of the previous point (data not shown)

Line 379 page 19: “among” is redundant

Line 386 page 19: data not shown should be reported, otherwise the sentence must be deleted

Line 400 page 19: also here, even if the topic is for my opinion more interesting, data about Epstein Barr virus, the data are not shown. I suggest to the authors to report these data in the method and result section. If the data will not be reported the sentence must be deleted

Author Response

The paper titled “Comprehensive gene analysis of IgG4-related ophthalmic disease using RNA sequencing”, reported on an interesting topic regarding RNA expression levels found by RNA-Seq in biopsy specimens of 3 patients diagnosed with IgG4-ROD compared to controls presenting with Mucosa-associated lymphoid tissue (MALT) lymphoma, reactive lymphoid hyperplasia (RLH), normal lacrimal gland tissue, and adjacent adipose tissue.

RNA-seq identified 35 upregulated genes including MMP12 and SPP1 in IgG4-ROD tissues when compared with all the controls, confirmed by real-time PCR and immunohistochemistry, that the authors considered having a role in the pathogenesis of Ig4-ROD and they could be considered as biomarkers.

The text and contents are understandable. However, there are some lack of information that should be added. Minor revision should be considered.

Minor English revision should be considered.

The title should be changed as this topic regarding only few patients: I suggest modifying as follow: Comprehensive gene analysis of IgG4-related ophthalmic disease using RNA sequencing: a preliminary case series study. 

Response

We thank the reviewer for the suggestion. Certainly, the results of RNA sequencing are preliminary because of the small number of cases. However, since validation by real-time PCR is performed in 60 cases, we would like to submit it with the original title.

The hypothesis of the possible role of a triggering infection as Epstein Barr infection should be well reported and the data of the possibile nifection including serology for each patient should be shown.

Response

We agree with the reviewer that Epstein Barr virus (EBV) infection as a triggering infection is important. Unfortunately, we do not have the EBV serological data of the patients who participated in this study.

Lines 88 and 89 page 2: please add Standard deviation (SD)

Response

Following the reviewer’s advice, we have added “±SD” in line 92.

Table 2 page 3: there is a mistake on the second line regarding Aex (male/female): IgG4-ROD is 13/30 or 13/17 and in MLAT lymphoma I 17/30 or 17/13?: please clarify

Response

We apologize for the typographical error and thank the reviewer for bringing it to our attention. We have corrected “Aex” to “Sex”. The figures are meant to be number of males/number of females. Since this presentation may cause confusion, we have changed to “Sex (male : female)”, “13 : 17” for IgG-ROD, and “17 : 13” for MALT lymphoma (Table 2).

Line 335 page 18: please delete the point before reference number 32

Response

We apologize for the typographical error. We have deleted the point before [32].

Lines 352-354 page 18: please delete the sentence When… as the data is not shown by the authors or the authors should decide to show the data.

Response

We have presented the result of our preliminary pathway analysis for genes upregulated in RLH relative to IgG4-ROD in a new supplementary table (Table S6). Therefore, we have preserved this sentence, and changed “(data not shown ) to “(Table S6)” (line 371).

Lines 385 -387 page 19: the same of the previous point (data not shown)

Response

As also commented by another reviewer, the preliminary finding and the hypothesis inferred may not be relevant to the present study. We have deleted this part (lines 398-408).

Line 379 page 19: “among” is redundant

Response

We have deleted “among” as advised (line 397).

Line 386 page 19: data not shown should be reported, otherwise the sentence must be deleted

Response

As mentioned above, this part has been deleted (lines 398-408).

Line 400 page 19: also here, even if the topic is for my opinion more interesting, data about Epstein Barr virus, the data are not shown. I suggest to the authors to report these data in the method and result section. If the data will not be reported the sentence must be deleted

Response

As mentioned above, while we think that analysis of EBV is important, we did not have full data for detailed investigation. We have therefore deleted the uncertain result together with the speculation (lines 417-420).

Reviewer 3 Report

IgG4-related disease (IgG4-RD) is a systemic disorder with lymphoplasmacytic fibrosis in multi-organs. Elevated serum IgG4 concentration is the hallmark of IgG4-RD. According to earlier studies, ~17% of IgG4-RD patients have ophthalmic manifestations, include lacrimal gland involvements, and orbital involvement, defined as IgG4-related ophthalmic disease (IgG4-ROD). Although the gene profiling of IgG4-RD has been reported, this is the first study focus the genes signature on IgG4-ROD. The authors found SPP1 and MMP12 higher expressed in IgG4-ROD samples, by confirmation by qPCR and immunohistochemistry.

Overall, this manuscript is largely interesting and significant. I have a few comments

  1. There are samples from 60 patients used for qPCR, does these samples include the samples used for RNA seq?
  2. In the previous study from the same group, they find the involvement of MAPK signalling pathway by microRNA profiling. How about MAPK signalling in this RNA seq analysis?
  3. For immunostaining, the validation of antibody specificity is missing.
  4. The Raw data of RNA seq are encouraged to be deposited to the public servers, like GEO in NCBI.

Author Response

IgG4-related disease (IgG4-RD) is a systemic disorder with lymphoplasmacytic fibrosis in multi-organs. Elevated serum IgG4 concentration is the hallmark of IgG4-RD. According to earlier studies, ~17% of IgG4-RD patients have ophthalmic manifestations, include lacrimal gland involvements, and orbital involvement, defined as IgG4-related ophthalmic disease (IgG4-ROD). Although the gene profiling of IgG4-RD has been reported, this is the first study focus the genes signature on IgG4-ROD. The authors found SPP1 and MMP12 higher expressed in IgG4-ROD samples, by confirmation by qPCR and immunohistochemistry.Overall, this manuscript is largely interesting and significant. I have a few comments

  1. There are samples from 60 patients used for qPCR, does these samples include the samples used for RNA seq?

Response

We thank the reviewer for the comment. Among the 30 samples of MALT lymphoma used in real-time PCR, the samples used in RNAseq were included. However, the 30 samples of IgG4-ROD used in PCR and the 3 samples used in RNAseq were different samples. We have added this information in the text (line 99-101)

  1. In the previous study from the same group, they find the involvement of MAPK signalling pathway by microRNA profiling. How about MAPK signalling in this RNA seq analysis?

Response

We appreciate the reviewer’s comment. Since we used a different pathway analysis tool in our previous study, there is no exact match between the pathways identified in the previous microRNA profiling and in the present RNA sequencing studies. However, in the present study, the DEG-based pathway analysis comparing IgG4-ROD with MALT lymphoma identified "FCERI mediated MAPK activation". (Fig. 3-c)

  1. For immunostaining, the validation of antibody specificity is missing.

Response

The manufacturer’s data sheet of mouse anti-human MMP12 (MAB919, R&D Systems, USA) shows that the antibody is specific. The URL for the catalog number MAB919 (given on page 4, line 132) is https://www.rndsystems.com/products/human-mmp-12-catalytic-domain-antibody-82902_mab919. The manufacturer’s data sheet for rabbit anti-human SPP1 (HPA027541, Atlas Antibodies, Sweden) does not contain information on specificity (https://www.atlasantibodies.com/api/print_datasheet/HPA027541.pdf), but since this antibody was used in many papers, the specificity is inferred.

  1. The Raw data of RNA seq are encouraged to be deposited to the public servers, like GEO in NCBI.

Response

We thank the reviewer for the suggestion, but we cannot register the raw data because we have not applied for genome registration to the Ethics Review Committee at the stage of research planning.

Reviewer 4 Report

This is a very comprehensive study of RNA expression in IgG4RD of the ocular adnexa compared with orbital fat, lacrimal gland, MALT lymphoma and Reactive lymphoid hyperplasia using RNA seq in 3 cases of each and real-time PCR for 30 IgG4RD and 30 MALT lymphomas.

The authors do not indicate if fresh tissue was used for RNA seq.

The authors demonstrate that the heatmap for IgG4RD and RLH are quite distinct with apparent converse expression for different genes for both, yet the principal component analysis plot shows some overlap.  How do the authors explain this?

Figure 8 legend does not indicate which figures are which tissues nor which antibody is used in each.

The authors state in line 324 that complement is depleted in IgG4-RD but in the article cited, C4 and C3 are only low in 20% and 19% of the cases studied, so this is the minority of cases rather than the majority.

Line 394 “we detected viral DNA in IgG4RD tissues.”  This needs to be expanded and explained as this is misleading.  The presence of scattered EBV+ B-cells and HHV6 and HHv7+ T-cells as a result of lymphoid infiltrates containing cells with latent infection with these viruses does not indicate pathogenicity of the virus as these are likely bystander effects.

The authors draw conclusions from their data which are purely hypothetical and not based on evidence eg the role of laminin-511 in pathogenesis of ocular adnexal IgG4-Rd.

Author Response

This is a very comprehensive study of RNA expression in IgG4RD of the ocular adnexa compared with orbital fat, lacrimal gland, MALT lymphoma and Reactive lymphoid hyperplasia using RNA seq in 3 cases of each and real-time PCR for 30 IgG4RD and 30 MALT lymphomas.

The authors do not indicate if fresh tissue was used for RNA seq.

Response

We thank the reviewer for the comment. After biopsy, each sample was cryopreserved at -80°C until analysis. We have added this information in the text (line 107-108).

The authors demonstrate that the heatmap for IgG4RD and RLH are quite distinct with apparent converse expression for different genes for both, yet the principal component analysis plot shows some overlap.  How do the authors explain this?

Response

We appreciate the reviewer’s comment. Principal component analysis (PCA) was performed using genes with p values less than 0.05 in ANOVA analysis. Therefore, the genes analyzed included some that did not differ between IgG4-ROD and RLH. On the other hand, only genes that differed between IgG4-ROD and RLH were used in the heat map analysis. This may have led to the results showing some overlap of IgG4-ROD and RLH in the PCA plot, but distinct difference between the two in the heatmap.

Figure 8 legend does not indicate which figures are which tissues nor which antibody is used in each.

Response

We apologize for incomplete descriptions in the legend. The upper row is MMP12 immunoreactivity and the lower row is SPP1 reactivity. From the left to the right column, IgG4-ROD, MALT lymphoma, RLH and lacrimal gland tissue sections are shown. We have added these explanations in the legend of Figure 8 (lines 315-317).

The authors state in line 324 that complement is depleted in IgG4-RD but in the article cited, C4 and C3 are only low in 20% and 19% of the cases studied, so this is the minority of cases rather than the majority.

Response

We apologize for the misleading expression. We intended to say that some patients with IgG4-related disease have hypocomplementemia. We have corrected the sentence accordingly (line 333).

Line 394 “we detected viral DNA in IgG4RD tissues.”  This needs to be expanded and explained as this is misleading.  The presence of scattered EBV+ B-cells and HHV6 and HHv7+ T-cells as a result of lymphoid infiltrates containing cells with latent infection with these viruses does not indicate pathogenicity of the virus as these are likely bystander effects.

Response

Again, we apologize for the misleading sentence. We have deleted this sentence to avoid misunderstanding (line 412).

The authors draw conclusions from their data which are purely hypothetical and not based on evidence eg the role of laminin-511 in pathogenesis of ocular adnexal IgG4-Rd.

Response

We appreciate the reviewer for the pertinent comment. We agree that conclusions drawn from data that are hypothetical are not appropriate. We have deleted such sentences including the role of laminin-511 (lines 398-408, lines 422-423).